# Characterizing the Mutational Landscape of Diffuse Large B-Cell Lymphoma in a Prospective Cohort of Mexican Patients

**DOI:** 10.3390/ijms25179328

**Published:** 2024-08-28

**Authors:** Myrna Candelaria, Dennis Cerrato-Izaguirre, Olga Gutierrez, Jose Diaz-Chavez, Alejandro Aviles, Alfonso Dueñas-Gonzalez, Luis Malpica

**Affiliations:** 1Clinical Research, The National Cancer Institute, Ciudad de Mexico 14080, Mexico; olgagut76@yahoo.com; 2Basic Research Division, Instituto Nacional de Cancerología, Ciudad de Mexico 14080, Mexico; dennis_cerrato@yahoo.com (D.C.-I.); josediaz030178@hotmail.com (J.D.-C.); 3Pathology Department, Instituto Nacional de Cancerología, Ciudad de Mexico 14080, Mexico; alejandroaviles2001@yahoo.com; 4Unidad de Investigación Biomédica en Cancer, Instituto Nacional de Cancerología, Ciudad de Mexico 14080, Mexico; alfonsoduenasg@yahoo.com; 5Department of Lymphoma/Myeloma, Division of Cancer Medicine, The University of Texas MD Anderson Cancer Center, Houston, TX 77030, USA; lemalpica@mdanderson.org

**Keywords:** mutational landscape, diffuse large B-cell lymphoma, prognostic, Latin America, genomics

## Abstract

Diffuse large B-cell lymphoma (DLBCL) is the most common B-cell malignancy worldwide. Molecular classifications have tried to improve cure rates. We prospectively examined and correlated the mutational landscape with the clinical features and outcomes of 185 Mexican patients (median age 59.3 years, 50% women) with newly diagnosed DLBCL. A customized panel of 79 genes was designed, based on previous international series. Most patients had ECOG performance status (PS) < 2 (69.2%), advanced-stage disease (72.4%), germinal-center phenotype (68.1%), and double-hit lymphomas (14.1%). One hundred and ten (59.5%) patients had at least one gene with driver mutations. The most common mutated genes were as follows: *TP53*, *EZH2*, *CREBBP*, *NOTCH1*, and *KMT2D*. The median follow-up was 42 months, and the 5-year relapse-free survival (RFS) and overall survival (OS) rates were 70% and 72%, respectively. In the multivariate analysis, both age > 50 years and ECOG PS > 2 were significantly associated with a worse OS. Our investigation did not reveal any discernible correlation between the presence of a specific mutation and survival. In conclusion, using a customized panel, we characterized the mutational landscape of a large cohort of Mexican DLBCL patients. These results need to be confirmed in further studies.

## 1. Introduction 

Lymphomas comprise a heterogeneous group of hematological malignancies classified according to their clinical, anatomopathological, and molecular features [1]. Diffuse large B-cell lymphoma (DLBCL) is the most common type of all aggressive lymphomas [2]. Presently, clinical prognostic scoring systems such as the international prognostic index (IPI) and the revised-IPI remain important tools with which to stratify patients based on their risk of early relapse and progression [3,4,5], yet refinement is warranted to incorporate novel predictive and prognostic biomarkers in DLBCL. 

The discovery of distinct DLBCL molecular subtypes based on cell of origin (COO) redefined the prognosis and treatment of DLBCL. The activated B-cell-like (ABC) and germinal center (GC) were associated with distinct response rates and survival outcomes to the standard of care chemoimmunotherapy as well as some targeted agents [6]. However, the aggressive clinical behavior observed in certain DLBCL subtypes, especially those exhibiting *MYC* rearrangement along with dual rearrangement with BCL2—defined as double-hit lymphomas [7]—underscored the necessity for a more comprehensive evaluation. 

In recent years, numerous genomic sequencing studies have primarily targeted recurrent mutations in individual genes, elucidating underlying oncogenic mechanisms [8]. Some researchers have undertaken a multiplatform analysis of structural genomic abnormalities, gene expression, or whole exome sequencing in DLBCL samples, aiming to establish a nosology of DLBCL based on shared genetic signatures [9,10,11]. This approach not only enhances our understanding of treatment response but also identifies therapeutic vulnerabilities, yielding promising outcomes. Notably, such an analysis has yet to be conducted in the Latin American population. Herein, this study aims to fill this gap by characterizing the mutational landscape and its correlation with clinical outcomes in a prospective cohort of Mexican patients with newly diagnosed DLBCL.

## 2. Results

### 2.1. Patient Characteristics

During the study period, 185 patients were included. The median age at diagnosis was 59.3 years (range 21 to 89) with a balanced sex distribution (female *n* = 93, 50.3%). Most patients had ECOG PS of <2 (*n* = 128, 69.2%), lacked B symptoms (*n* = 131, 70.8%), and had advanced-stage disease (stage III–IV, *n* = 134, 72.4%). Bone marrow involvement was confirmed in 28 of 75 tested patients (37%). Using the Hans classification, most had a GC (*n* = 126, 68.1%) phenotype. Forty-six (24.8%) cases overexpressed *MYC* and BCL2 and were considered double expressors. FISH analysis was performed only in double expressors and 26 were classified as double-hit lymphoma.

### 2.2. Mutational Landscape of Mexican Patients with DLCBL

Targeted sequencing was achieved successfully with an average coverage of 98.4% and a sequencing depth greater than 500X. A total of 1,642 somatic variants were found in 44 out of the 79 genes comprising our custom panel. Driver and passenger mutations were identified, with 110 patients (59.4%) having at least one gene with driver mutations (Figure 1a). Passenger mutations were excluded and only driver mutations were compared with clinical characteristics and used for the subsequent analysis. 

The clinical features of patients with or without driver mutations are presented in Table 1. A total of 272 driver mutations were identified in 23 different genes. A detailed description of the location, type of mutation, and protein change seen for every patient is shown in Appendix A. Missense variants were the most common type of mutation (72.16%), followed by stop-gain variant (21.25%) mutation (Figure 1b). *TP53* (32%*), EZH2* (31%*), CREBBP* (23%*)*, *NOTCH1* (17%*),* and *KMT2D* (15%*)* were the five most frequent mutated genes (Figure 1c). An enrichment analysis was performed in comparison to the Reactome R-HAS-212436 Gene Set for the Generic Transcription Pathway. Within the 23 genes with driver mutations, different pathways were found to be enriched with mutations, especially the NOTCH4 intracellular domain transcription regulation pathway which includes the genes *CREBBP*, *NOTCH2*, *EP300,* and *NOTCH1* (Figure 1d).

### 2.3. Correlations between Clinical Features and Mutational Profiles

We compared the frequency of driver mutations with the patients’ clinical features (Figure 2). Mutations affecting the *KMT2D*, *EZH2*, and *BCOR* genes were most frequent in female patients (*p*-value = 0.025, *p*-value = 0.027, *p*-value = 0.042, respectively). In contrast, *MYD88* was more frequent with male patients (*p*-value = 0.030). An advanced clinical stage (III-IV) was significantly associated with *TP53* (*p*-value = 0.009), *EZH2* (*p*-value = 0.011), and *DDX3X* mutations (*p*-value = 0.040). The GCB DLBCL subtype was associated with mutations in *EZH2* (*p*-value = 0.003) and *CREBBP* (*p*-value = 0.030) genes. On the other hand, non-GCB DLBCL was associated with *MYD88* mutations (*p*-value = 0.030). The presence of a bulky mass was associated with mutations in *EZH2* (*p*-value = 0.014) and *KMT2D* (*p*-value = 0.029) genes. The presence of B symptoms was common in *EZH2* mutated DLBCL (*p*-value = 0.030). The presence of ≤1 extranodal disease site was associated with *EZH2* (*p*-value = 0.003), whereas >2 extranodal sites were associated with *MYD88* mutations (*p*-value = 0.017). No associations were found between the different driver mutations and the IPI score or response to chemoimmunotherapy. The complete analysis results for the different driver mutations and clinical features are presented in Appendix A.

### 2.4. Survival Analysis

All patients were treated with R-CHOP. The response rates achieved after multi-agent chemotherapy were complete responses in 129 patients (69.7%), partial responses in 10 patients (5.4%), and stable disease in three patients (1.6%); 26 patients (14%) had progressive disease while on treatment. Seventeen patients (9.3%) were not evaluable for response due to early therapy discontinuation related to intolerable toxicity and/or death from the disease. In this analysis, 35 cases were older than 65 years and were treated with standard doses of R-miniCHOP. The overall response rate was not different between patients < 65 vs. older: 79.6% and 78.4% (*p* = 0.925).

With a median follow-up of 42 months (range 10 to 100 months), the 5-year RFS and OS rates were 70% and 72%, respectively. The RFS and OS rates were not different between patients with and without specific driver mutations (Table 1) and are shown in Figure 3.

### 2.5. Impact of Clinical and Molecular Features on Survival

We performed an exploratory analysis evaluating the impact of patients’ clinical features and mutational profile in RFS and OS. In the univariate analysis, ECOG PS > 2 (Hazard Ratio [HR]: 1.80; confidence interval [CI] 95%: 1.20–2.60, *p*-value = 0.003), the presence of bulky disease (HR: 2.60; CI 95%: 1.40–4.90, *p*-value = 0.002), advanced-stage disease (HR: 1.40; CI 95%: 1.00–2.00, *p*-value = 0.034), and a high-risk IPI score (HR: 1.40; CI 95%: 1.10–1.80, *p*-value = 0.007) were associated with inferior RFS, whereas age > 50 years at diagnosis (HR: 1.00; CI 95%: 1.00–1.10, *p*-value = 0.001), ECOG PS > 2 (HR: 1.70; CI 95%: 1.20–22.50, *p*-value = 0.004), advanced-stage disease (HR: 1.50; CI 95%, *p*-value 0.021), and a high-risk IPI score (HR: 1.40; CI 95%: 1.10–1.80, *p*-value = 0.006) were associated with inferior OS.

Also, we found the presence of ≤2 gene with driver mutations as a positive prognostic factor associated with better OS (HR: 0.31; CI 95%: 0.12–0.80, *p*-value = 0.01). No specific driver mutations were found to influence RFS and OS. Interestingly, the two cases that carried *ARID1A* mutations died at 12 and 36 months of follow-up and had inferior OS (HR: 4.30; CI 95%: 1.00–18.00, *p*-value = 0.043). The complete univariate analysis for all studied variables can be found in Appendix A.

The multivariate analysis was adjusted for age >50 years at diagnosis, ECOG PS > 2, and the presence of two or more genes with driver mutations had a negative impact on OS, whereas the use of none had a statistically significant impact in RFS (Table 2).

## 3. Discussion

In this study, we systematically analyzed the mutational landscape of newly diagnosed DLBCL patients from Mexico, with a particular emphasis on genes previously identified as clinically significant in other populations. We built a gene panel able to identify 272 driver mutations from 23 different genes. In our cohort, 110 patients (59.4%) had at least one driver mutation, with the most frequent mutations being *TP53* (32%), *EZH2* (31%), *CREBBP* (23%), *NOTCH1* (17%), and *KMT2D* (15%). We found age > 50 years at diagnosis, ECOG > 2, and the presence of two or more mutations to be negative factors impacting survival. Moreover, the two patients with *ARID1A* mutations died before the median OS. To our knowledge, this is the first study evaluating the mutational landscape of Hispanic DLBCL patients using in-house technology. 

To date, the COO classification of DLBCL remains the most widely used way to classify DLBCL subtypes, with up to 20% remaining unclassifiable. Although non-GCB or ABC subtypes have the poorer prognosis, noninterventional study has been able to overcome the outcomes of these patients [12,13]. Recently, the POLARIX trial reported a potential benefit in progression-free survival of patients with ABC DLBCL subtype managed with polatuzumab-R-CHP [14]. However, a benefit in OS remains to be proven with longer a follow-up. 

A deeper and more detailed analysis using genomic, transcriptome, and epigenetic data has been performed by different groups, refining the classification of DLBCL into clusters with impact in clinical outcomes [9,10]. As an example, Wang et al. describes cluster 1 as genes regulating pathways involved with DNA methylation/demethylation, histone methyltransferase, and protein methyltransferase activity; cluster 2 is accompanied by *B2M*, *CD70,* and *MEF2B* mutations, which are related with DNA damage repair, and cytokine-mediated and B-cell activated immune signaling [15]. In our study, we devised a tailored panel comprising genes consistently reported by prior studies. However, we could not discern specific subgroup classifications.

In our study, mutations in *TP53* were the most frequent molecular feature in our DLBCL cohort. This finding differs from previous reports including those performed in non-Caucasian populations. Mamgain et al. described *NOTCH1* (16.1%) and *ARID1A* (12.9%) mutations as the most common molecular features of DLBCL in India [16]. The frequency of *NOTCH1* mutations was similar in our study, but we cannot compare with other populations in Latin America, since to our knowledge, this is the first study evaluating the genomic landscape in such population. A study including an Asian population [17] found IGH fusion (69%), PIM1 (33%), MYD88 (29%), BCL2 (29%), TP53 (29%), CD79B (25%) and KMT2D (24%) as the most common mutations. Patients with *TP53* mutations were more likely to have primary refractory disease (87.0% vs 50.0%, *p*-value = 0.009). In our series, *TP53* mutations had no impact on relapse rates or survival rates. *ARID1A* has been implicated in the pathogenesis of Epstein–Barr virus (EBV)-associated DLBCL [18]. In our cohort, the two cases harboring *ARID1A* were not tested for presence of EBV proteins. 

The methyltransferase activity of *EZH2* towards lysine 27 on histone H3 (H3K27) and non-histone proteins dysregulates and alters chromatin compaction, protein complex recruitment, and transcriptional regulation. At least three stable isogenic mutants (*EZH2*^Y641F^, *EZH2*^A677G^, *EZH2^H^*^689A/F667I^) have been described [19,20,21]. These mutants have impacts on pathways involved in differentiation, migration, and cellular proliferation. In our study, mutations in *EZH2* were common in females and in patients with advanced clinical stage, bulky mass, B symptoms, and <1 extranodal site. *EZH2* was also most common in GC type DLBCL, as it has been previously described in other populations [19,22,23,24,25].

*MYD88* has been widely evaluated in DLBCL [26,27,28,29,30,31,32,33,34,35]. In our study, we found *MYD88* mutations in 14 cases. To date, mutations in *MYD88* have been implicated in primary breast DLBCL [20], primary renal lymphoma [26], central nervous system DLBCL [27,28,34], and primary testicular lymphoma [33]. The mutational signature of primary immuno-privileged site DLBCL has been described as heterogeneous, in comparison with non-specified DLBC [36]. Although we had a limited number of patients, the presence of *MYD88* mutations were significantly more frequent in males, in the non-GC type, and in those with extensive extranodal disease. *MYD88* and *TP53* genes have a controversial role in prognosis and survival. When compared to clinical prognostic systems (i.e., IPI), mutations in these genes have been associated with a worse overall response rate and shorter survival [30]. However, in our study, we did not find an association between the presence of *MYD88* and *TP53* mutations, and response and/or survival rates. Further studies are needed to identify if this lack of association is due to the Hispanic ethnicity or any other molecular features.

The classifications of DLBCL, divided by a group from the National Institutes of Health from the USA [9] into 4 subtypes, differ phenotypically by gene expression signatures and responses to treatment, with inferior outcomes in the MCD and N1 subtypes and favorable survival rates in the EZB and the BN2 groups. However, the majority of both ABC and GBC cases were unclassified. With the same aim of classification, Chapuy et al. [10] performed a comprehensive genetic analysis, including low-frequency alterations, recurrent mutations, somatic copy number alterations, and structural variants to integrate with these combined methodologies five clusters with prognostic implications. When we try to correlate the findings of both groups, there is a similarity between these molecular subtypes, in particular MCD with C5, BN2 with C1, and EZB with C3. In contrast to these authors, our study is limited because we performed gene sequencing, but we did not use other molecular techniques, in particular those directed at evaluating structural chromosomal changes. Despite this, we also documented that EZH2 and CREBB mutations were more frequent in CGB lymphomas, while MYD88 variants were present in non-GCB lymphomas. In the same direction, TP53 and EZH2 mutations were associated with an advanced clinical stage, suggesting they may be associated with a more aggressive disease. However, this is a theory that needs to be confirmed. 

In this study, our multivariate analysis showed inferior OS in patients with advanced age at diagnosis, ECOG > 2, and the presence of two or more mutated genes. In our cohort, nonclinical and/or molecular factor had a clear impact on RFS. Interestingly, only the presence of ≥2 mutations impacted OS, but not RFS; this may be related to the impact of the multiple mutations in other pathways implicated not only in lymphoma but also in metabolism or other comorbidities influencing OS, as has been described in other tumors [37,38].

A previous study showed *ANKRD26*, *BRCA2*, *MYD88*, and *NOTCH1* as factors associated with high relapse rates in patients with DLBCL [22]. In our study, the two patients that harbored *ARDI1A* had a shorter OS. Thus, we cannot speculate on the possible negative impact of this gene in our patient cohort.

## 4. Material and Methods

### 4.1. Study Design

This is a prospective, non-randomized, observational study of consecutive patients with newly diagnosed DLBCL, evaluated and managed at the National Cancer Institute (INCan) in Mexico City, Mexico, between January 2015 and December 2017, and with a last follow-up in January 2023. The inclusion criteria were patients aged 18 years and older with a histopathological diagnosis of DLBCL, who were treatment-naïve and fit for chemoimmunotherapy. We excluded patients with active hepatitis B, C, and/or HIV infection and central nervous system lymphoma. Exclusion criteria included patients whose tumor sample was insufficient for sequencing (n = 6). The study protocol was approved by the IRB (registration number CEI/966/15). All patients provided informed consent.

### 4.2. Study Variables

The clinical variables analyzed were age, gender, IPI score, the presence of bulky disease (defined as >7 cm), B symptoms, clinical stage based on the modified Ann Arbor staging [39], serum albumin, lactate dehydrogenase (LDH), and performance status (PS) as evaluated by the ECOG PS score [40]. The histopathological variables analyzed included the following: GC versus non-GC, as assessed by Hans nomogram [41], and *BCL2*, *BLC6*, and *MYC* expression and/or rearrangement. Lymphomas co-expressing *MYC* and *BCL2* or *BLC6* were considered to be double expressors. Those harboring *MYC* rearrangement by fluorescence in situ hybridization (FISH) along with *BCL-2* and/or *BCL-6* rearrangement were classified as double-hit or triple-hit lymphomas. All patients were treated with 6 cycles of rituximab, cyclophosphamide, doxorubicin, vincristine, and prednisone (R-CHOP) regimen, as previously described [42]. The clinical response was evaluated by positron emission tomography and computed tomography (PET/CT) scanning at baseline and end of treatment using Deauville’s criteria [43]. After therapy completion, all treated patients were followed with clinical, laboratory, and computed tomography (CT) scanning for up to eight years.

### 4.3. Tissue Sample Preparation 

Tumor DNA was extracted from formalin-fixed paraffin-embedded (FFPE) tissue block of the primary neoplasm using the kit Quick-DNA/RNA FFPE kit (Zymo Research, Orange, CA, USA), and quantified using fluorometry in Qubit 3 equipment (Thermo Fisher Scientific, Waltham, NA, USA). DNA purity was evaluated by spectrophotometry in NanoDrop 1000 (Thermo Fisher Scientific). DNA integrity was analyzed on a 1% agarose gel. For uracil removal, 20 ng of DNA was pretreated with 1 µL (1 U) uracil DNA glycosylase (Thermo Fisher Scientific) plus 20 µL of water and incubated (37 °C × 2 min and 50 °C × 10 min); 10 ng of each sample was used for the libraries. For analyses, a comprehensive literature review was undertaken to identify publications featuring genomic analyses that categorized DLBCL patients into distinct clusters [9,10,11]. After the literature review, a custom panel was meticulously designed, comprising genes. This was consistently and repeatedly reported by these authors. This custom panel (6026 oligos) was used for the sequencing of coding regions and splicing sites of 79 genes (Appendix A) associated with the progression, diagnosis, and treatment response of lymphomas according to the literature, using the Ion AmpliSeq Chef library kit. Pools of eight samples were loaded onto Chip Ion 550 (Thermo Fisher Scientific). Then, the libraries were quantified by qPCR with the TaqMan RNase P detection kit (Thermo Fisher Scientific) and sequenced using the Ion GeneStudio S5 Prime System platform v. 5.18.

### 4.4. Bioinformatic Tools 

Data analysis was carried out using Ion Torrent Suite Browser version 5.0 and Ion Reporter version 5.18. Variant calling was performed with Torrent Variant Caller. Single-nucleotide variants (SNVs) and small insertions and deletions (InDels) were annotated with Cancer Genome Interpreter (CGI) (v. 2018), as were databases such as ClinVar (201706) [44], COSMIC (v86) [45], PolyPhen (v2.2.2) [46], SIFT (v5.2.2) [47], FATHMM (v2.1) [48], gnomAD (r2.0.1) [49], and the Variant Effect Predictor (v93.2) [50]. Synonymous mutations and mutations annotated in non-coding regions were filtered out. The biological and clinical relevance of SNVs and InDels were identified using the Cancer Genome Interpreter (CGI) web interphase [51]. For predicting driver mutations with CGI, a tissue-specific model (large B-Cell lymphoma) was selected. CGI uses a machine learning algorithm named BoostDM to annotate the clinical and biological relevance of the somatic variants with a BoostDM score of 0.5 and an accuracy for predicting driver mutations (F50-Score) of above 0.9 [52].

### 4.5. Statistical Analysis

Demographic characteristics are reported using descriptive statistics. Patients were divided into those with at least one gene with driver mutations and those without one. Differences in clinical characteristics were evaluated in each group. Associations between genes with driver mutations and clinical characteristics such as age; sex; COO, as assessed by a Hans nomogram; the presence of extranodal disease; bulky mass; IPI score; and the response to chemotherapy were assessed based on the frequency of SNVs and InDels. Patients were grouped according to their clinical characteristics, and then the frequency of the variants between the groups was compared using Fisher’s exact test. *p*-values were adjusted using the Benjamini–Hochberg procedure. Unadjusted *p*-values < 0.05 were considered as significant.

Survival curves were estimated using the Kaplan–Meier method. Overall survival (OS) was measured from the time of diagnosis until the time of death or last follow-up. Relapse-free survival (RFS) was measured from the time of diagnosis until the time of relapse. Cox proportional-hazard regression analysis was used to identify the variables predicting OS, measured from the time of diagnosis until the time of death or last follow-up. *p*-values < 0.05 were considered statistically significant.

All statistical analyses were performed using R software (v4.2.2) (https://cran.r-project.org, accessed on 23 May 2023) and SPSS software (v25) (IBM, Corp., Armonk, NY, USA).

## 5. Conclusions

Our study contributes to the expanding body of literature on DLBCL in Hispanic patients by elucidating the prevalence of *TP53*, *CREBBP*, and *NOTCH1* mutations, among other genes within our cohort, diverging from established trends observed in diverse populations. Our observations suggest associations between select mutations (e.g., EZH2 and MYD88) and clinical phenotypes, but their prognostic significance remains to be proven in a larger cohort of Hispanic patients. Furthermore, our multivariate analysis identified advanced age, compromised performance status, and polygenic mutational burden as determinants of inferior OS, consistent with established prognostic parameters. However, the prognostic impact of specific mutations, such as *ARID1A*, necessitates further scrutiny given our study’s constrained sample size. A principal limitation of our investigation regards the absence of germline DNA, owing to our reliance on FFPE tissue blocks for genetic profiling. Nonetheless, this study’s robustness is underscored by the inclusion of a sizable, prospectively followed cohort of Mexican DLBCL patients, offering valuable insights into the molecular underpinnings and prognostic determinants of this clinically heterogeneous disease.

## Figures and Tables

**Figure 1 ijms-25-09328-f001:**
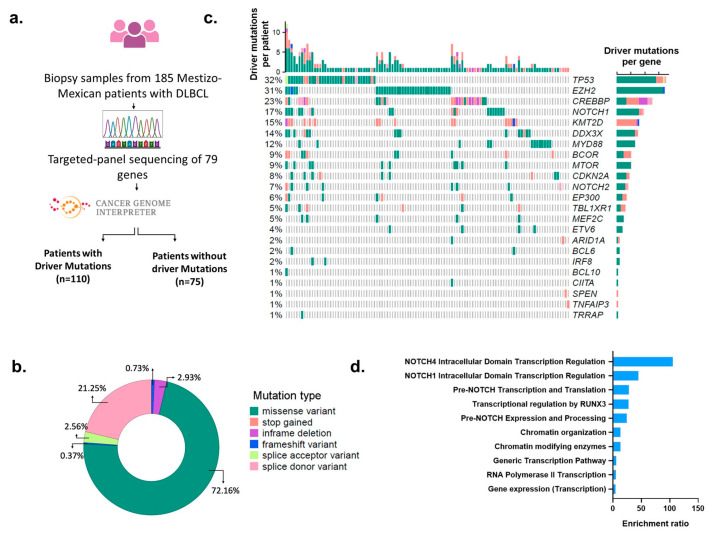
Mutational landscape of patients with diffuse large B-cell lymphoma. (**a**) The classification of patients according to the presence of driver mutations. (**b**) The frequency of mutation types found in the sequenced samples. (**c**) OncoPrint (complexheatmap v.2.20.0) depicts the frequency of driver mutations per patient and gene found in the sequenced samples. (**d**) Enrichment analysis, performed with the Reactome database (v.88), displaying the molecular pathways affected by driver mutations in patients with diffuse large B-cell lymphoma (DLCBL).

**Figure 2 ijms-25-09328-f002:**
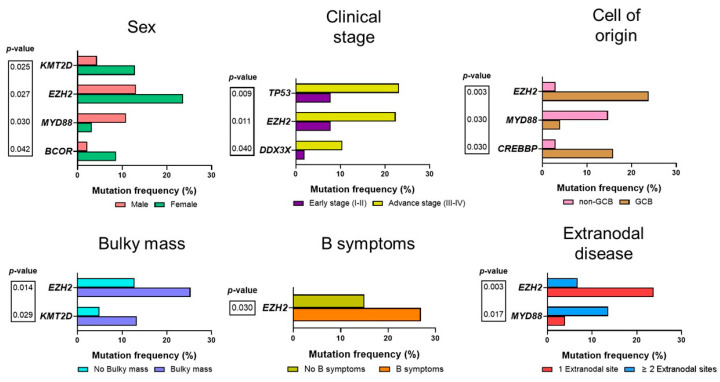
Associations between clinical features and the mutational landscape of patients with diffuse large B-cell lymphoma. Each bar plot depicts the frequency of genes found to be significantly different on each clinical characteristic. *p*-value is according to the Fisher exact test. A complete list of the analysis between the clinical features and the genes with driver mutations can be found in Appendix A.

**Figure 3 ijms-25-09328-f003:**
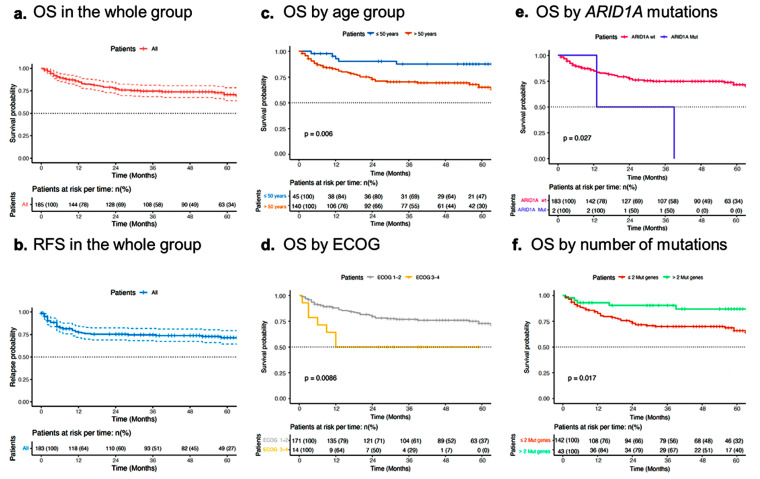
Overall survival of the 185 patients sequenced. (**a**) Disease-free survival of the 183 patients who received chemotherapy treatment. (**b**) Overall survival (OS) in the whole groups. (**c**) OS by age group (< vs. ≥50 years). (**d**) OS by EGOG group (≤2 vs. ≥3). (**e**) OS by ARID1 mutations. (**f**) OS by number of genes mutated (0–1 vs. >2).

**Table 1 ijms-25-09328-t001:** Clinical characteristics of the whole group and comparison with and without driver mutations.

All Patients			Driver Mutations				
			None		Present		
Characteristic	N = 185	% (100)	N = 75	% (100)	N = 110	% (100)	*p*
Female/male	93/92	50.3/49.7	37/38	49.3/50.7	56/54	50.9/49.1	0.44
ECOG: 0–1 2 3 4	128 43 12 2	69.2 23.2 6.5 1.1	58 13 4 0	77.4 17.3 5.3 -	70 30 8 2	63.7 27.2 7.3 1.8	0.19
B symptoms No Yes	131 54	70.8 29.2	57 18	76.0 24.0	74 36	67.2 32.8	0.27
Bulky No Yes	102 83	55.1 44.9	45 30	60.0 40.0	56 54	51.0 49.0	0.33
Clinical stage: I II III IV	18 33 21 113	9.7 17.8 11.4 61.1	9 19 7 40	12.0 25.3 9.3 53.4	9 14 14 73	8.1 12.8 12.8 66.3	0.09
Extranodal disease None 1 site 2 sites >3 sites	66 67 20 32	36 37 11 16	30 24 10 11	40.0 32.0 13.4 14.6	39 43 10 18	35.4 39.2 9.1 16.3	0.44
IPI score *: Low Intermediate–low Intermediate–high High	45 40 35 65	24.4 21.6 19 35	21 20 12 22	28.0 26.6 16.0 29.4	24 20 23 43	21.8 18.2 20.9 39.1	0.24
Cell of origin: GC Non-GC	126 59	68.1 31.9	49 26	65.3 34.7	77 33	70.0 30.0	0.24
Response to treatment: Complete Partial Stable Progressive Not evaluable	129 10 3 26 17	69.7 5.4 1.6 14.0 9.3	56 4 1 8 6	74.8 5.3 1.3 10.6 8.0	73 6 2 18 11	66.5 5.4 1.8 16.3 10.0	0.77
Median ± SD RFS (months)	39.12 + 27.97		41.31 + 27.94		37.63 + 28.09		0.44
Median ± SD OS (months)	41.99 + 26.47		42.52 + 27.17		41.63 + 26.09		0.83

* IPI = international prognostic index. GC = germinal center. Non-GC = non-germinal center. RFS = relapse-free survival. OS = overall survival.

**Table 2 ijms-25-09328-t002:** Influence of clinical features and genes with driver mutations on OS and RFS.

	Multivariate Analysis for OS			Multivariate Analysis for RFS		
	HR	IC 95%	*p*-Value	HR	IC 95%	*p*-Value
Age	1.04	1.01–1.06	**0.002**	1.02	0.91–1.06	0.26
ECOG	1.50	0.99–2.28	**0.021**	1.32	0.96–1.98	0.055
Bulky mass	1.33	0.70–2.53	0.568	1.34	0.71–2.52	0.449
Clinical stage	1.47	0.99–2.16	0.061	1.44	0.97–2.13	0.066
IPI score	0.88	0.60–1.28	0.702	0.94	0.65–1.37	0.635
*ARID1A*	7.98	1.91–35.24	0.006	5.17	1.19–2.59	
Number of driver mutations	0.33	0.13–0.83	**0.016**	0.34	0.13–1.12	0.086

## Data Availability

None previously reported results have been published.

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
