# Peer review of "Characterizing the Mutational Landscape of Diffuse Large B-Cell Lymphoma in a Prospective Cohort of Mexican Patients"

_ijms, 2024, doi:10.3390/ijms25179328_

Round 1

Reviewer 1 Report

Comments and Suggestions for Authors

L68: FISH was only performed in double expressors? Please specify

Table 1: there is an extra line after the word Median in the RFS

You use DFS or RFS indistinctly. Please, unify.

Table 2: I consider more interesting inserting the median and IQ range values of the variable with hazard ratio and the p value than the b, SE and z values.

L216: you should include as one of the possible explanations of the lack of association between TP53 and MYD88 and clinical outcomes the sample size. Also, the type of TP53 mutation could explain these differences.

L221: ARID1A instead of “ARDI1A”

In general, the nomenclature of genes must be corrected with the indistinct use of italics.

I miss a comparison of the mutational spectrum of your series with the published literature in other populations. You mention India, but very few about Caucasian, where there is the most of the literature.

In addition, I recommend specifying the type of the TP53 mutation. At least, in which proportion the mutations affected the DNA binding domain or generated a truncated protein.

Author Response

1. L68: FISH was only performed in double expressors? Please specify.

RE: Yes, FISH analysis was done only in double expressors.  (see lines 68 &69)

2. Table 1: there is an extra line after the word Median in the RFS

It was removed.

3. You use DFS or RFS indistinctly. Please, unify.

It was unified as RFS: line 25, 98,  129, 139,  143, 149,155, table 2, 220, 296 & figure 3

4. Table 2: I consider more interesting inserting the median and IQ range values of the variable with hazard ratio and the p value than the b, SE and z values. 

It was done. See lines 160-163

Multivariate analysis for OS

Multivariate analysis for RFS

HR

IC 95%

p-value

HR

IC 95%

p-value

Age

1.04

1.01-1.06

0.002

1.02

0.91-1.06

0.26

ECOG

1.50

0.99-2.28

0.021

1.32

0.96-1.98

0.055

Bulky mass

1.33

0.70-2.53

0.568

1.34

0.71-2.52

0.449

Clinical stage

1.47

0.99-2.16

0.061

1.44

0.97-2.13

0.066

IPI score

0.88

0.60-1.28

0.702

0.94

0.65-1.37

0.635

ARID1A

7.98

1.91-35.24

0.006

5.17

1.19-2.59

Number of  driver mutations

0.33

0.13-0.83

0.016

0.34

0.13-1.12

0.086

4. L216: you should include as one of the possible explanations of the lack of association between TP53 and MYD88 and clinical outcomes the sample size. Also, the type of TP53 mutation could explain these differences.

We didn´t found an association between TP53 and MYD88 mutations. As you can see in the following table, only 2 patients had both mutations.

We agree, it is very important to describe the site and kind of mutations. Therefore we detailed every mutation, including TP53 mutatios in each patient in a supplementary table 2.

6. L221: ARID1A instead of “ARDI1A”

IT was corrected.

7. In general, the nomenclature of genes must be corrected with the indistinct use of italics.

It was reviewed and corrected.

8. I miss a comparison of the mutational spectrum of your series with the published literature in other populations. You mention India, but very few about Caucasian, where there is the most of the literature.

You are completely right. Since the studies done by Chapuy et al & Schmidt et al included additional approaches that we couldn´t do, we compared our results and pointed out this limitation, as follows in lines 221-236.

The classification of DLBCL by a group of the National Institutes of Health from USA (9) in 4 subtypes and differed phenotypically by gene expression signatures and response to treatment with inferior outcome in the MCD and N1 subtypes and favorable survival in the EZB and the BN2 groups.However, the majority of both ABC and GBC cases were unclassified. With the same aim of classification, Chapuy et. al (10) did a comprehensive genetic analysis, including low-frequency alterations, recurrent mutations, somatic copy number alterations and structural variants to integrate with these combined methodologies 5 clusters with prognostic implications. When we try to correlate the findings of both groups there is a similarity between these molecular subtypes, in particular MCD with C5, BN2 with C1 and EZB with C3.  In contrast with these authors, Our study is limited because we did gene sequencing, but we didn´t use other  molecular techniques, in particular directed to evaluate structural chromosomal changes. Despite this, we also documented that EZH2 and CREBB mutations were more frequent in CGB lymphomas, as well as MYD88 were present in nonGCB lymphomas. In the same direction TP53 & EZH2 mutations were associated with advanced clinical stage, suggesting it may be associated with a more aggressive disease. However, this is a theory that need to be confirmed. 

9. In addition, I recommend specifying the type of the TP53 mutation. At least, in which proportion the mutations affected the DNA binding domain or generated a truncated protein.

Thank you very much for this commentary. The supplementary table 2 detailes the gene mutations found in every patient, including the aminoacid change in the protein.

In lines 85-87 it is referred as: A detailed description of the location, type of mutation, and amino acid change for every patient is shown in Supplementary Table 2.

Reviewer 2 Report

Comments and Suggestions for Authors

The article “Characterizing the Mutational Landscape of Diffuse Large 2 B-cell Lymphoma in a Prospective Cohort of Mexican Patients” by Candelaria M et al. investigated the mutational landscape in association with clinical features and prognosis in 185 Mexican patients with DLBCL those were all treated by the conventional six cycles of R-CHOP, using a customized panel for 79 genes. The authors showed the mutational profiles in their cohort and did not reveal any significant correlation between a specific mutation of driver genes and survival. This is the first report examining the mutational landscape in Mexican patients with DLBCL, and the authors showed the prognostic impact of polygenic mutational burden in addition to advanced age and compromised performance status. However, the lack of evidence for the association with mutation patterns and prognosis may be due to insufficient patients, necessitating further analysis and considerations. Careful interpretation of the result is needed.

1.     The number of mutations appeared to be associated with OS but not DFS. Then, what did the number of mutations directly impact to cause the shorter OS?

2.     Although the authors focused on ARID1A mutations, the mutation was seen only in two patients. I don’t think this study can sufficiently investigate the prognostic impact of ARID1A mutation.

3.     Some mutations, such as those of EZH2 and MYD88, seemed to be associated with gender. Is this incidental or scientifically meaningful?

4.     This study is prospective, so the authors should examine and discuss the prognostic impact of the relative dose intensity of R-CHOP.

5.     Authors may examine the prognostic impact of gene mutations along with previously reported genetic classification, such as those published by Wright GW (Cancer Cell 2020) and by Chapuy B (Nat Med 2018), in the current cohort to see if the genetic mutation indeed lacks a prognostic role in their cohort

6.     The authors may investigate the mutational signature in association with prognosis.

7.     The prognostic impact of each gene mutation may also be examined in two separate groups, i.e., the GCB group and the non-GCB group.

8.     The authors need to see the possible confounding bias between age and some specific mutation.

Author Response

  1. The number of mutations appeared to be associated with OS but not RFS. Then, what did the number of mutations directly impact to cause the shorter OS?

We analyzed the impact on OS and RFS of 1, 2, 3 or more mutations in this proyect. Howerver, only the presence of  > 2 mutations had an impact on OS. The absence of impact on RFS may be due that gene mutations may  affect different pathways associated  not only with lymphoma, but also with other systems and comorbidities. 

This information is on lines 149-150 as: Also, we found the presence of < 2 gene with driver mutations as a positive prognostic factor associated to better OS (HR: 0.31; CI 95 %: 0.12 -0.80, p-value = 0.01).

& lines: 222- 225: Also, we found the presence of < 2 gene with driver mutations as a positive prognostic factor associated to better OS (HR: 0.31; CI 95 %: 0.12 -0.80, p-value = 0.01).

The impact of >2 mutations on OS is illustrated on figure 3.

  1. Although the authors focused on ARID1A mutations, the mutation was seen only in two patients. I don’t think this study can sufficiently investigate the prognostic impact of ARID1A mutation.

You are completely right. We described mutations at ARID1A gene because it was statistically significant, although on lines 310-312 we emphasize that a deeper research is required:  “such as ARID1A, necessitates further scrutiny given our study's constrained sample size”

3. Some mutations, such as those of EZH2 and MYD88, seemed to be associated with gender. Is this incidental or scientifically meaningful?

Although in DLBLC the gender has not been described as a prognostic factor, as in other lymphoproliferative diseases (Ex: Hodgkin Lymphoma), we considered of clinical importance to analyze the association of gene mutations with clinical parameters at diagnosis, including: Age, IPI, ECOG, gender. We documented that  EZH2 and MYD88 mutations were more frequent within female, and males respectively.

If the presence of EZH2 and MYD88 mutations has a clinical impact on the evolution between men and women is not known, and would require to be evaluated  in a future proyect.

  1. This study is prospective, so the authors should examine and discuss the prognostic impact of the relative dose intensity of R-CHOP.

Thank you very much for the recommendation of this analysis. This was a prospective study including patients with a mean age 59.3 years. An inclusion criteria was patients without previous treatment, candidate to receive full  & standard doses R-CHOP. Although the range of age was from 21 to 89 years, 35 cases were older than 65 that required a reduction on intensity of the schema and received R-miniCHOP, as it is  internationally recommended. However, the overall response rate was not different between boths subgroups. This not statiscally significant difference was added in lines 129130, as:

In this analysis, 35 cases were older than 65 years and were treated with standard doses of R-miniCHOP. The overall response rate was not different between patients < 65 vs older: 79.6 % & 78.4 % (p=0.925).

  1. Authors may examine the prognostic impact of gene mutations along with previously reported genetic classification, such as those published by Wright GW (Cancer Cell 2020) and by Chapuy B (Nat Med 2018), in the current cohort to see if the genetic mutation indeed lacks a prognostic role in their cohort

You are completely right. Since the studies done by Chapuy et al & Schmidt et al included additional approaches that we couldn´t do, we compared our results and pointed out this limitation, as follows in lines 221-236.

The classification of DLBCL by a group of the National Institutes of Health from USA (9) in 4 subtypes and differed phenotypically by gene expression signatures and response to treatment with inferior outcome in the MCD and N1 subtypes and favorable survival in the EZB and the BN2 groups.However, the majority of both ABC and GBC cases were unclassified. With the same aim of classification, Chapuy et. al (10) did a comprehensive genetic analysis, including low-frequency alterations, recurrent mutations, somatic copy number alterations and structural variants to integrate with these combined methodologies 5 clusters with prognostic implications. When we try to correlate the findings of both groups there is a similarity between these molecular subtypes, in particular MCD with C5, BN2 with C1 and EZB with C3.  In contrast with these authors, Our study is limited because we did gene sequencing, but we didn´t use other  molecular techniques, in particular directed to evaluate structural chromosomal changes. Despite this, we also documented that EZH2 and CREBB mutations were more frequent in CGB lymphomas, as well as MYD88 were present in nonGCB lymphomas. In the same direction TP53 & EZH2 mutations were associated with advanced clinical stage, suggesting it may be associated with a more aggressive disease. However, this is a theory that need to be confirmed. 

6. The authors may investigate the mutational signature in association with prognosis.

Although we did an exhaustive statistical analysis, we did not identify a prognosis signature. Therefore, our results are descriptive.

7.- The prognostic impact of each gene mutation may also be examined in two separate groups, i.e., the GCB group and the non-GCB group.

Thank you very much for this comment. In the Material and Methods section, lines 292-295 we described that all mutations and clinical factors were analyzed bin GCB group and non-GCB group,  as follows:

Associations between genes with driver mutations and clinical characteristics such as age, sex, COO by Hans nomogram, presence of extranodal disease, bulky mass, IPI score, and response to chemotherapy were assessed with the frequency of SNVs and InDels

The mutations that were found more frequently and statistically significant in the GCB group and the non-GCB group, were illustrated in figure 2: EZH2 and CREBB mutations within the GCB group; In contrast, MYD88 mutations in non-GCB group.

8. The authors need to see the possible confounding bias between age and some specific mutation.

As shown in table 2, the multivariate analysis included age, ECOG, the presence of bulky mass, clinical stage and IPI score.

Round 2

Reviewer 2 Report

Comments and Suggestions for Authors

I have no additional requirement.